# COVID-19 Immunization Rates in Patients with Inflammatory Bowel Disease Worldwide: A Systematic Review and Meta-Analysis

**DOI:** 10.3390/vaccines11101523

**Published:** 2023-09-25

**Authors:** Francesco Paolo Bianchi, Rossella Donghia, Rossella Tatoli, Caterina Bonfiglio

**Affiliations:** 1Epidemiology Unit, Bari Policlinico General Hospital, 70124 Bari, Italy; 2National Institute of Gastroenterology, IRCCS S. De Bellis, Research Hospital, 70013 Castellana Grotte, Italy; rossella.donghia@irccsdebellis.it (R.D.); rossella.tatoli@irccsdebellis.it (R.T.); catia.bonfiglio@irccsdebellis.it (C.B.)

**Keywords:** Crohn’s disease, ulcerative colitis, vaccine compliance, SARS-CoV-2, gastroenterology, biological drugs, immunosuppressive therapy, immunization rates

## Abstract

Individuals with Inflammatory Bowel Disease (IBD) are characterized by an increased vulnerability to complications stemming from infectious diseases. While these patients do not inherently face a heightened risk of SARS-CoV-2 infection compared to the general population, their vulnerability to severe COVID-19 complications and subsequent hospitalization is notably increased. The objective of our study is to quantitatively assess the global coverage of COVID-19 vaccination among individuals with IBD, achieved through a comprehensive meta-analysis and systematic review. Thirteen studies were systematically selected from scientific articles available in the MEDLINE/PubMed, ISI Web of Knowledge, and Scopus databases, spanning from 1 January 2021 to 25 July 2023. The pooled prevalence of COVID-19 vaccine uptake was estimated at 72% (95%CI = 59–83%) for at least one dose, 81% (95%CI = 68–91%) for the complete vaccination regimen, and 71% (95%CI = 46–91%) for the third dose. Analysis of the determinants influencing vaccination uptake revealed several significant associations. These encompassed Caucasian ethnicity, female sex, absence of immunosuppressive therapy, advanced age, prior receipt of the anti-influenza vaccine, absence of a history of COVID-19 infection, and the provision of advice from gastroenterologists, all linked to improved compliance. Our study underscores a noteworthy yet not entirely optimal COVID-19 vaccination coverage among individuals with IBD. A multifaceted approach is warranted to enhance vaccination rates. Within this context, the role of gastroenterologists extends beyond direct patient care, encompassing a pivotal responsibility in preventing complications stemming from post-infectious diseases.

## 1. Introduction

Inflammatory Bowel Disease (IBD) encompasses chronic inflammation of the gastrointestinal tract, with Crohn’s disease and ulcerative colitis being the most prevalent forms [1]. While IBD is commonly associated with young adults, it can affect individuals at any age, with about 25% of patients experiencing symptoms before age 20 [2]. The incidence of IBD, including early-onset cases, is on the rise globally, with recent studies highlighting significant growth in very early-onset IBD in specific regions [3]. In the United States, an estimated 3.1 million adults, comprising 1.3% of the population, have been diagnosed with IBD [1].

Although the exact cause of IBD remains uncertain, it is widely believed that an abnormal or dysregulated immune response targeted at the intestinal microbiota, coupled with genetic susceptibility, contributes to its development. While individuals with IBD exhibit disrupted intestinal microbiota, it is unclear whether this dysbiosis is the cause or a result of the disease [2]. Despite ongoing research to elucidate this complex relationship, it is evident that patients with IBD face an increased risk of complications related to infectious diseases. Several factors contribute to this heightened infectious risk, including the use of immunosuppressive drugs to manage inflammation, the higher probability of having other chronic health conditions (cardiovascular disease, respiratory disease, cancer, arthritis, kidney disease, liver disease, etc.) compared to subjects without IBD, and older age, which is associated with immunocompromised states [4,5,6].

Regarding the COVID-19 pandemic, patients with IBD are not at a higher risk of SARS-CoV-2 infection than the general population [7]. However, they are more susceptible to severe complications from COVID-19. A 2022 multicenter study reported older age (aOR = 1.04; 95%CI = 1.01–1.02), having two or more comorbidities (aOR = 2.9; 95%CI = 1.1–7.8), using systemic corticosteroids (aOR = 6.9; 95%CI = 2.3–20.5), and using sulfasalazine or 5-aminosalicylate (aOR = 3.1; 95%CI = 1.3–7.7) as risk factors for severe COVID-19 in patients with IBD [8]. Conversely, treatment with a tumor necrosis factor antagonist was not associated with severe COVID-19 risk (aOR = 0.9; 95%CI = 0.4–2.2). In a study by Xu F et al. [9], COVID-19 hospitalization rates and outcomes were compared between US patients with and without IBD, suggesting that older adults with IBD may have a higher risk of hospitalization. Finally, Bezzio et al. [10] found a higher risk of COVID-19 complications in patients with active IBD, older age, and comorbidities.

In light of the increased susceptibility to severe COVID-19, international gastroenterology scientific societies and public health institutions recommend COVID-19 vaccination for patients with IBD and immunocompromised individuals [11,12]. Although phase III trials for COVID-19 vaccines did not include IBD patients or those on immunosuppressive therapies, phase IV studies have shown that the vaccines are highly effective, immunogenic, and safe for this subgroup population [4,13,14,15,16].

Despite the potential benefits of vaccination, high rates of vaccine hesitancy have been reported among patients with IBD. Surveys in the United States conducted in 2022 demonstrated hesitancy rates ranging from 12 to 36% among patients with IBD, with higher hesitancy observed among younger subjects [17,18]. Vaccine hesitancy in this population is a well-known phenomenon, with several studies reporting low vaccine coverage for other vaccine-preventable diseases [19,20,21]. Various reasons contribute to this hesitancy, including concerns about vaccine safety, perceptions of vaccine necessity or effectiveness, lack of healthcare professional recommendations, limited knowledge about vaccines, access barriers, costs, and conflicting advice [19,20,21].

Our study aims to estimate COVID-19 vaccine coverage (VC) among patients with IBD worldwide through a meta-analysis and comprehensive systematic review. Additionally, we seek to identify determinants of vaccine uptake and explore strategies to enhance vaccination attitudes. In particular, we will emphasize the proposed strategies to assess the promotion of vaccination among these patients.

## 2. Materials and Methods

The systematic review protocol was established in accordance with the Preferred Reporting Items for Systematic Reviews and Meta-Analyses (PRISMA) checklist [22]. The protocol was registered in the International Prospective Register of Systematic Reviews (PROSPERO) under the reference acknowledgment number CRD42023453662. The review question was formulated using the Population, Intervention, Comparison, and Outcome (PICO) framework, with the question being “What is the COVID-19 vaccine coverage (O) among patients with IBD (P) worldwide, and what are the related determinants (C and I) of vaccine uptake?”.

### 2.1. Search Strategy, Selection Criteria, and Data Extraction

The Scopus, MEDLINE/PubMed, and ISI Web of Knowledge databases were systematically searched for relevant articles. The search included research articles, brief reports, letters, and editorials published between 1 January 2021, and 25 July 2023. The search strategy used the following terms: (“Inflammatory bowel disease” OR IBD OR crohn OR “ulcerative colitis”) AND (“vaccine status” OR “vaccine coverage” OR uptake OR “immunization status” OR hesitan* OR compliance OR attitude) AND (COVID* OR coronavirus OR sars-cov-2). Only studies in English with full text were included. Abstracts without full text, reviews, meta-analyses, papers not reporting epidemiological data, clinical trials, and studies unrelated to the purpose of this review (e.g., vaccine hesitancy, seroprevalence, safety) were excluded. If needed, the authors of the studies were contacted to gather additional information. The list of papers was screened independently by two reviewers based on their titles and/or abstracts, and the predefined inclusion and exclusion criteria were applied. Any discrepancies were recorded and resolved through consensus.

The extracted data encompassed various elements, such as the year of publication, sample size, number of vaccinated patients, age of patients, country of study, potential determinants of vaccine uptake, and strategies employed to enhance VC.

### 2.2. Quality Assessment

The methodological quality of the selected studies was evaluated using the Newcastle-Ottawa Scale (NOS), adapted for assessing cross-sectional studies, as proposed by Modesti P.A. et al. [23]. The NOS consists of seven categories, examines three quality aspects (selection, comparability, and outcome/exposure), and assigns scores ranging from 0 to 10. A high-quality study was defined by a NOS score between 7 and 10, an intermediate-quality study between 4 and 6, and a low-quality study between 0 and 3. Two independent researchers conducted the risk of bias assessment for each study, and any discrepancies were recorded and resolved through consensus.

### 2.3. Main Outcome and Pooled Analysis

A meta-analysis was performed to estimate the COVID-19 VC among patients with IBD, and a sub-analysis was conducted to estimate the VC based on geographical areas. For the meta-analysis, the pooled proportion was calculated using the Freeman-Tukey double arcsine transformation to stabilize variances, and the DerSimonian-Laird weights were used for random effects models.

The odds ratios (ORs) and 95% confidence intervals (CIs) were selected as general outcome variables to assess the relationship between COVID-19 vaccine uptake and various analyzed determinants, such as sex, disease, race, previous influenza shot, etc. The ORs and standard errors (SEs) were calculated from the 95% CIs, and an additional logarithmic transformation was performed to stabilize the variance and normalize the distribution. The OR in the meta-analysis was calculated using the inverse variance and DerSimonian-Laird weights for random effects models.

Heterogeneity was estimated using the inverse-variance random-effects model. A *p*-value < 0.05 was considered statistically significant for heterogeneity, and the I index was estimated and interpreted as follows:0% to 40%: not significant heterogeneity;30% to 60%: moderate heterogeneity;50% to 90%: substantial heterogeneity;75% to 100%: considerable heterogeneity.

Three sensitivity analyses were conducted to assess stability, including:sub-analysis considering only high-quality studiessub-analysis based on study sample size (419+ vs. <419 patients); the median value of the sample of the included studies was used to define this cut-off.exclusion of one study at a time to evaluate potential distortions and ensure the robustness of the conclusions.Statistical analysis was performed using STATA MP18 software.

The determinants of COVID-19 vaccine uptake were collected from all available studies, and the respective findings were compared, with particular attention to the evidence presented in several of the included papers.

## 3. Results

### 3.1. Identification of Relevant Studies

The article selection process was conducted following the PRISMA guidance [22], and the flow chart depicting this process is presented in Figure 1. In total, 330 articles were identified from three databases: 77 from the ISI Web of Knowledge, 159 from Scopus, and 94 from MEDLINE/PubMed. After removing duplicate articles across the databases, 17 studies were considered eligible. Among the identified articles, two studies were excluded because more recent and comprehensive articles, already included in the systematic review, evaluated the same phenomenon in the same sample. Furthermore, two studies were excluded due to the unavailability of the full text. As a result, 13 studies were considered eligible and met the inclusion criteria [24,25,26,27,28,29,30,31,32,33,34,35,36] (Table 1), while 111 studies were excluded for not meeting the specified criteria. Two studies were from the US, two from China, two from the UK, two from Italy, one from Poland, one from Denmark, one from Canada, one from Australia, and one from Kuwait (Table 1, Figure 2).

### 3.2. Quality Assessment

The NOS was appropriately applied to assess the quality of the included studies, and it was found that 38.5% of the studies were of high quality (Table 1).

### 3.3. Pooled Analysis

The pooled prevalence of COVID-19 vaccine uptake estimated on 113,191 patients, considering at least one dose, was found to be 71.7% (95%CI: 59.4–82.6%; I^2^ = 99.4%; *p*-value for heterogeneity < 0.0001). When considering only high-quality articles, the pooled prevalence increased to 77.9% (95%CI: 67.9–86.7%; I^2^ = 98.8%; *p* < 0.0001). Comparing the studies based on sample size, the pooled vaccine uptake prevalence was 82.7% (95%CI: 69.6–92.6%; I^2^ = 99.4%; *p* < 0.0001) in studies with a sample size larger than 418 patients and 63.1% (95%CI: 48.1–77.0%; I^2^ = 97.2%; *p* < 0.0001) in studies with a smaller sample size. The *p*-value for the heterogeneity test between these sub-groups was 0.043 (Figure 3). Additionally, conducting sensitivity analysis by excluding one study at a time revealed no significant distortion from a specific paper.

The prevalence of vaccine uptake for the completed vaccination series estimated on 110,795 patients was 80.6% (95%CI: 68.1–90.7%; I^2^ = 99.1%; *p*-value for heterogeneity < 0.0001). When considering only high-quality articles, the pooled prevalence increased to 83.8% (95%CI: 83.6–84.1%; I^2^ = -; *p* = -). Comparing the studies based on sample size, the pooled vaccine coverage prevalence was 88.5% (95%CI: 81.9–93.8%; I^2^ = -; *p* = -) in studies with a sample size larger than 418 patients and 63.7% (95%CI: 59.6–67.8%; I^2^ = -; *p* = -) in studies with a smaller sample size. The *p*-value for the heterogeneity test between these sub-groups was <0.0001 (Figure 4). Additionally, conducting sensitivity analysis by excluding one study at a time did not reveal any significant distortions from a specific paper.

The prevalence of vaccine uptake for the third dose estimated on 1432 patients was 71.4% (95%CI: 46.4–90.8%; I^2^ = 98.9%; *p*-value for heterogeneity < 0.0001). Due to the small number of eligible studies investigating this topic, sensitivity analyses were not performed.

In a subanalysis by continents, the pooled prevalence of VC for at least one vaccine dose was as follows: 78.4% (95%CI: 61.1–91.6%; I^2^ = -; *p* = -) in North America (three studies), 54.8% (95%CI: 37.1–71.8%; I^2^ = -; *p* = -) in Asia (three studies), 72.0% (95% CI: 40.5–94.7%; I^2^ = -; *p* = -) in Europe (four studies), and 93.2% (95%CI: 90.5–95.1%) in Oceania (one study). The *p*-value for the test of heterogeneity between subgroups was <0.0001. Sensitivity analyses did not reveal any significant distortion.

Estimates of OR values when comparing vaccine uptake for several determinants are presented in Table 2. Sensitivity analyses, where applicable, were conducted to account for the limited number of studies that investigated specific determinants. These sensitivity analyses did not indicate any significant distortion in the findings.

### 3.4. Determinants of Vaccine Uptake and Strategies to Raise Vaccine Coverage

Most studies have identified critical reasons for negative attitudes towards vaccination, including a lack of information about vaccines, concerns about vaccine safety, and fear of potential adverse events [24,25,27,35]. In particular, several authors have noted that patients express fears that vaccination could exacerbate existing conditions such as IBD and other chronic illnesses [24,26,27,33]. For instance, Kwon H.J. et al. [26] discovered a negative association between previous IBD-related surgeries and COVID-19 vaccination, suggesting that individuals with a history of severe IBD might exhibit more hesitancy towards vaccination. Furthermore, patients using certain medications appear to prefer avoiding vaccination [24,28,30]. Specifically, Cao Y. et al. [25] observed that IBD patients on biologics might be more inclined to avoid COVID-19 vaccinations, while those on 5-ASA treatment showed a greater willingness to be vaccinated. In contrast, Kwon H.J. et al. [26] reported a positive association between biological use and COVID-19 vaccination, with no significant relationship observed for other immunosuppressive medications.

Minor factors contributing to a negative attitude toward the vaccine included monthly household income [28], a lack of available time for vaccination [32], a history of prior COVID-19 infection [27,28,35], and the presence of a household member aged over 65 years [34]. Family members and friends played a role in influencing vaccination compliance [27], along with faith in the effectiveness of vaccines [27] and a self-perceived higher risk for COVID-19 [28,33]. Patients frequently cited concerns about their safety and protecting their family members as primary motivations for getting vaccinated [27]. Lodyga M. et al. [27] highlighted that individuals who relied heavily on mass media and social media for information exhibited higher vaccine hesitancy. In contrast, those with higher levels of education and information from credible scientific sources tended to have a more favorable acceptance of vaccines. Moreover, a significant determinant of vaccination compliance was the history of receiving prior vaccinations, particularly the influenza vaccine [27,28,33,34].

Regarding gender, greater levels of compliance were noted among females [26,28,35], and various studies [25,26,27,28,30,31,32,35] indicated reduced hesitancy among older individuals. Social determinants have also emerged as significant factors. Not being part of an ethnic minority [26,28,31], having a higher economic status [27,31], and residing in an urban area [27,28] are linked to enhanced vaccine compliance.

In this context, various strategies have been highlighted to enhance immunization coverage. Numerous authors [24,25,26,27,28,29,30,32,34,35] underscored the pivotal role of healthcare providers, particularly gastroenterologists. Trust in healthcare professionals as reliable sources of information regarding COVID-19 vaccine safety emerges as a robust predictor of vaccine acceptance. It is crucial for healthcare workers to proactively address misconceptions among IBD patients, such as the unfounded fear that vaccination might trigger an IBD flare [33]. Kwon H.J. et al. [26] established that patients with more frequent clinic visits over 12 months were more likely to receive the COVID-19 vaccine. Clinic visits provide valuable opportunities for patients to seek clarification about the vaccine and its potential impact on their condition. Some studies suggested that patients with frequent clinic visits might be in an acute phase of their disease, heightening their perception of the risk and susceptibility to COVID-19 and motivating them to opt for vaccination [26,28]. As highlighted by Lodyga M. et al. [27], healthcare professionals must engage in comprehensive discussions with patients, weighing the benefits and risks of vaccination and arriving at informed decisions tailored to individual circumstances. In alignment with this, healthcare providers should remain consistently updated about the latest scientific advancements [27]. Sciberass N. et al. [29] proposed the involvement of a multidisciplinary team, including specialist nurses and psychological support, to alleviate anxiety, promote adherence to safe measures, and minimize the likelihood of IBD flares.

Considerable emphasis is placed on effectively communicating evidence-based information about vaccine safety and efficacy to mitigate vaccine hesitancy among patients with IBD [35]. As a result, public health strategies should prioritize educating healthcare providers and the general public about the potential side effects of COVID-19 and the continuously evolving safety data concerning vaccines within this demographic. It is recommended to leverage mass media, social media, and the internet to disseminate evidence-based information about the vaccine and COVID-19 [35]. Education should take precedence, encompassing more than just knowledge acquired from television screens or computers. This trust safeguards against misinformation and empowers individuals to fully harness medicine’s benefits [27].

Ultimately, numerous studies [24,25,33,35] have centered around individuals with IBD in the context of vaccine trials, leaving clinicians with the challenge of advising vaccination for these patients without substantial evidence regarding both effectiveness and safety, particularly during the early stages of vaccination campaigns. Establishing an optimal approach to incorporating these patients into upcoming vaccine studies instead of automatically excluding them emerges as a critical agenda item for the scientific community and policymakers.

## 4. Discussion

Our meta-analysis estimated the VC among patients with IBD worldwide to be 72% (95%CI = 59–83%) for at least one dose, 81% (95%CI = 68–91%) for the complete vaccination regimen, and 71% (95%CI = 46–91%) for the third dose. While these figures might initially appear promising, the 20% of unvaccinated individuals remain a cause for concern. Indeed, although patients with IBD do not demonstrate a higher susceptibility to infection than the general population [7], the scientific literature underscores the increased risk of complications and hospitalization within this specific subgroup. This risk escalates, particularly for those undergoing treatment, the elderly, or those with multiple underlying health conditions [8,9,10]. As a result, ensuring a near-100% vaccination uptake rate becomes imperative.

Inequality in the adoption of vaccines has been noted in the continent-based sub-analysis, displaying greater adoption rates in Western nations and lower rates in Asia. This incongruity is expected, as scientific literature underscores reduced VC in less developed countries compared to their counterparts in more developed regions. This contrast primarily stems from socioeconomic factors and obstacles to vaccine accessibility [37,38].

An analysis of the assessed determinants of vaccination uptake unveiled several factors associated with improved compliance. These included Caucasian ethnicity, female gender, absence of immunosuppressive therapy, prior receipt of the anti-influenza vaccine, and no history of COVID-19 infection, findings that are corroborated by our systematic review. Social disparities acting as barriers to vaccination uptake, not exclusively for anti-COVID-19 vaccines, have been established as determinants by other evidence within the literature [39,40,41]. The significance of receiving a previous anti-influenza vaccination as a determinant of vaccination adherence is well documented and reported in the literature for various sub-groups [42,43,44]. Our study also reveals a contrasting pattern in gender-based compliance among individuals with IBD. Specifically, it reveals that women with IBD exhibit higher compliance than men. This observation diverges from existing scientific literature, where a 2022 meta-analysis indicated a lower tendency for women to opt for COVID-19 vaccination than men on a global scale [45]. Additional research is essential to shed more light on this intriguing aspect.

Regarding age, the available evidence suggests that older individuals display greater adherence to vaccination [25,26,27,28,30,31,32,35]. This inclination can be attributed to the increased susceptibility of advanced age to COVID-19 complications, which is frequently accompanied by an increased prevalence of other underlying health conditions. Consequently, older individuals tend to exhibit a heightened awareness of their elevated risk profile.

It is noteworthy that there is evidence of lower vaccine uptake among patients undergoing steroid or biological therapy; our systematic review also elucidates the reasons behind this hesitancy, with patients expressing fears about vaccine effectiveness and the potential for severe side effects, such as exacerbation of the underlying disease [24,25,26,27,33]. Indeed, the safety profile of vaccination in individuals receiving immunosuppressive treatment is comparable to that in healthy subjects; similarly, while slightly reduced in immunocompromised individuals, the effectiveness remains sufficiently high to safeguard vulnerable patients [4,13,14,15,16]. In particular, a multicenter Italian study conducted in 2022 interviewed 809 vaccinated IBD patients. Approximately 45% of these patients reported experiencing at least one side effect, with 10% reporting such effects after the first dose, 15% after the second dose, and 19% after receiving both doses. All adverse events were of mild intensity and had a short duration, lasting only a few days. This study revealed statistically significant associations between adverse events and factors such as female sex, younger age, seroconversion, and comorbidities [46].

Advanced education and reputable scientific sources significantly influence the perspectives of this population. Trust in the scientific community has previously been recognized as a crucial factor affecting vaccination adherence within the broader population [47,48], and it similarly emerges as a pivotal element within this subgroup. The dissemination of misinformation through mainstream and social media contributes to vaccine skepticism, a phenomenon our systematic review validated among individuals with IBD [27]. Indeed, there is a clear call to promote vaccination using traditional and contemporary information channels, aiming to counteract misinformation and establish institutional communication grounded in scientific evidence [35].

The role of a gastroenterologist in advocating for vaccinations among patients with IBD is paramount. These specialized medical practitioners comprehensively understand the unique healthcare needs of individuals with IBD. They can offer tailored guidance regarding the safety profile of vaccines and provide reassurances about their efficacy within this at-risk population. By leveraging their expertise, gastroenterologists can make informed recommendations that address the specific considerations of each patient’s disease status, treatment plan, and overall health. This proactive approach prioritizes the well-being of IBD patients and contributes to broader public health goals by promoting vaccination coverage in this vulnerable population.

The primary limitation of this meta-analysis was the substantial heterogeneity observed among the studies, as evidenced by the I^2^ values. However, the adoption of a random-effects analysis helped mitigate this potential bias. A significant constraint lay in the overall low-quality rating attributed to the included studies, given that most comprised letters. Moreover, a noteworthy proportion of these studies were authored by gastroenterologists, which a broader public health perspective might have supplemented to enhance their reliability. The relatively small number of studies included in the meta-analysis underscores a knowledge gap in the scientific literature. Moreover, most of the research originated in developed nations, with a notable absence of studies from Africa or South America. Consequently, a substantial portion of the global landscape remains unrepresented in the literature. This critical gap necessitates a heightened emphasis on international research on vaccine uptake within this vulnerable population. Such an approach is imperative for future repetitions of meta-analyses and systematic reviews, intending to yield more robust and comprehensive outcomes. Another limitation of this study is the inability to stratify the number of vaccinated individuals by vaccine type, given that the majority of included studies do not provide this information. Consequently, drawing conclusions on this topic is challenging. However, our review and meta-analysis boasted a significant strength in the considerable sample size derived from aggregating selected studies. This augmentation positively impacted the statistical analysis and provided a more comprehensive understanding of the COVID-19 vaccine uptake among individuals with IBD. Moreover, a notable aspect of our study was the calculation of ORs for several determinants, a feature not previously explored in the existing literature.

In summary, our study showed high but not wholly satisfactory VCs among patients with IBD. A multifactorial approach is needed to raise VC, considering that fighting vaccine resistance is a harsh and slow process. Health education should be provided to improve the community’s willingness, especially for those with lower levels of education [49]. Government and Public Health institutions should guarantee clear and unambiguous communication to express the vaccine’s risk/benefit ratio for vulnerable populations, especially focusing on younger individuals; it is appropriate for public health institutions to ensure proper institutional and scientific communication, especially on social networks [50]. Patients with IBD have been excluded from COVID-19 vaccine clinical trials because of ethical concerns and the potential for liability [24,25,33,35]. So, there is a lack of evidence from phase III studies supporting the safety and efficacy of vaccines in this specific population, leaving clinicians with the challenge of advising vaccination for these patients without substantial evidence, particularly during the early stages of vaccination campaigns.

Collaboration among diverse healthcare professionals who treat individuals with IBD, encompassing general practitioners, public health experts, and specialists, is strongly recommended. In this context, the pivotal role of gastroenterologists takes on added significance, as their responsibilities should extend beyond direct patient care to encompass the prevention of post-infectious disease complications. The exacerbation and decompensation of IBD contribute to heightened hospitalizations and an increased workload for specialists, culminating in a deteriorating clinical scenario and additional community costs. Equally crucial is the active participation of nosocomial facilities in immunization strategies, proactively offering vaccination prophylaxis to chronic patients. This approach has proven successful, as documented in various studies within the literature [51,52]. Integrating the mitigation of infectious risks into the treatment pathways of patients with IBD is paramount. By diminishing the likelihood of complications, we enhance patient management, elevate their quality of life, and foster more favorable responses to treatments and care. Prioritizing COVID-19 vaccination (among other vaccinations) within this susceptible population can substantially reduce hospitalizations, mortality rates, and associated complications.

In conclusion, the significance of the COVID-19 vaccination for patients with IBD cannot be overstated. As our understanding of the virus and its potential complications continues to evolve, prioritizing vaccination within this vulnerable population emerges as a pivotal strategy for safeguarding their health and well-being. Comprehensive vaccination coverage necessitates a multifaceted approach involving close collaboration among healthcare professionals, including gastroenterologists, general practitioners, and public health officials. By fostering a unified effort, we can establish tailored immunization protocols that account for the unique medical considerations of individuals with IBD. The ramifications of such strategic endeavors are far-reaching, extending beyond the individual patient to have substantial implications for public health. Enhanced vaccination rates among this cohort translate into decreased hospitalizations, reduced disease transmission, and a lighter burden on healthcare systems. The proactive approach to vaccination shields patients from severe COVID-19 outcomes and contributes to the broader goal of curbing the pandemic’s impact. As we forge ahead, prioritizing COVID-19 vaccination for individuals with IBD remains a pivotal step toward a healthier, more resilient society. 

## Figures and Tables

**Figure 1 vaccines-11-01523-f001:**
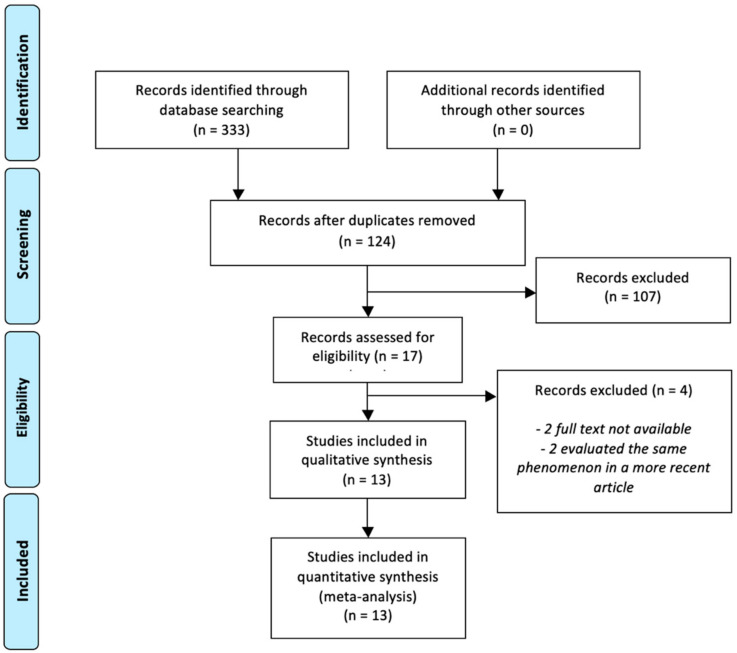
Flow-chart of the bibliographic research.

**Figure 2 vaccines-11-01523-f002:**
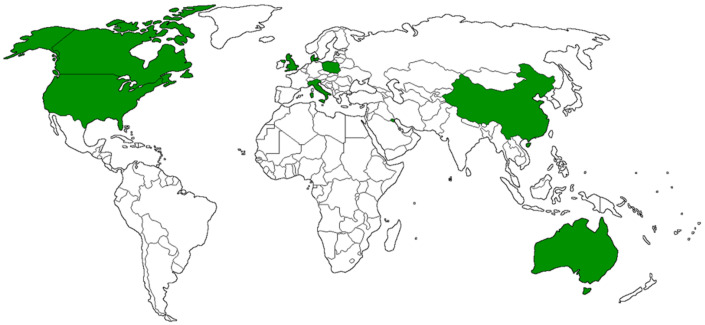
Cartogram of the studies included in the systematic review.

**Figure 3 vaccines-11-01523-f003:**
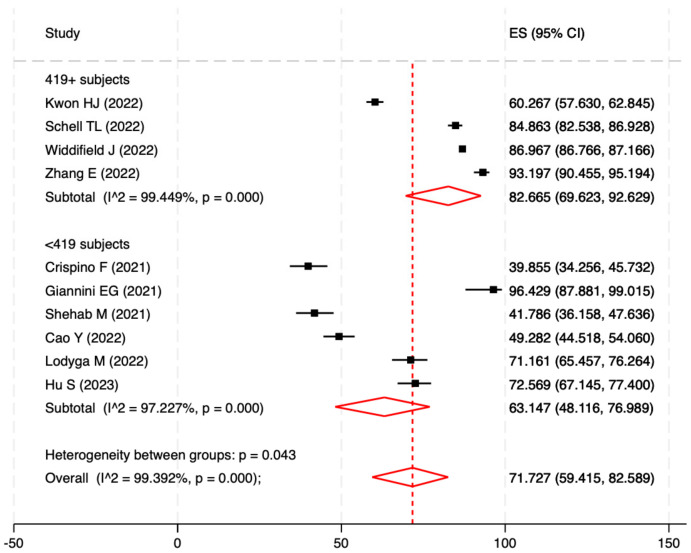
Forest plot of the pooled prevalence of vaccine coverage for at least one dose of the COVID-19 vaccine as determined by the eligible studies’ sample (419+ patients vs. <419 patients).

**Figure 4 vaccines-11-01523-f004:**
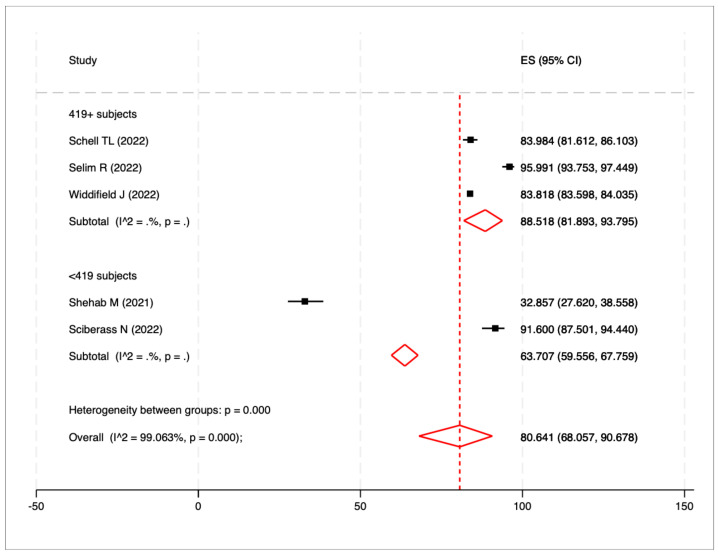
Forest plot of the pooled prevalence of vaccine coverage for the COVID-19 complete vaccination series as determined by the eligible studies’ sample (419+ patients vs. <419 patients).

**Table 1 vaccines-11-01523-t001:** Characteristics of the selected studies included in the meta-analysis and systematic review.

Author	Year	Quality	Sample	n. of Vaccinated Patients with at Least One Dose	n. of Vaccinated Patients with Completed Vaccination Series	n. of Vaccinated Patients with the Third Dose	Study Period	Country	Authorized Vaccines	Population
Hu S.	2023	h	288	209			September 2021–December 2021	China	Sinopharm BIBP, CoronaVac, Convidecia, Zifivax, Minhai, Chinese Academy of Medical Sciences, V-01	Adults
Cao Y.	2022	h	418	206			January 2021–July 2021	China	Sinopharm BIBP, CoronaVac, Convidecia, Zifivax, Minhai, Chinese Academy of Medical Sciences, V-01	Adults
Kwon H.J.	2022	m	1349	813			March 2020–October 2021	US	Pfizer–BioNTech, Moderna, Janssen, Novavax	Adults
Lodyga M.	2022	h	267	190			September 2021–November 2021	Poland	Pfizer–BioNTech, Moderna, Janssen, Novavax, Oxford-Astrazeneca, VLA2001, Sanofi-GSK	Adults
Schell T.L. *	2022	h	1024	869	860		November 2020–April 2021	US	Pfizer–BioNTech, Moderna, Janssen, Novavax	Adults
Sciberass N.	2022	m	250		229	214/246	March 2020–June 2022	Denmark	Pfizer–BioNTech, Moderna, Janssen, Novavax, Oxford-Astrazeneca, VLA2001, Sanofi-GSK	Adults
Selim R.	2022	m	449		431	205	April 2020–January 2021	UK	Pfizer–BioNTech, Moderna, Janssen, Novavax, Oxford-Astrazeneca, VLA2001	Adults
Wellens J.	2022	m	733			580	April 2020–unknown	UK	Pfizer–BioNTech, Moderna, Janssen, Novavax, Oxford-Astrazeneca, VLA2001	Adults
Widdifield J.	2022	h	108,792	94,613	91,187		December 2020–July 2021	Canada	Pfizer–BioNTech, Moderna, Janssen, Novavax, Oxford-Astrazeneca, CoVLP	Adults
Zhang E.	2022	h	441	411			October 2021–November 2021	Australia	Pfizer–BioNTech, Moderna, Janssen, Novavax, Oxford-Astrazeneca, COVAX-19	Adults
Crispino F. *	2021	m	276	110			April 2021	Italy	Pfizer–BioNTech, Moderna, Janssen, Novavax, Oxford-Astrazeneca, VLA2001	Adults
Giannini E.G. *	2021	l	56	54			Unknown	Italy	Pfizer–BioNTech, Moderna, Janssen, Novavax, Oxford-Astrazeneca, VLA2001	Adults
Shehab M.	2021	m	280	117	92		June 2021–October 2021	Kuwait	Pfizer–BioNTech, Moderna, Janssen, Sinopharm BIBP, Oxford-Astrazeneca, CoronaVac, Covaxin	Adults

* letter.

**Table 2 vaccines-11-01523-t002:** Estimation of the Odds Ratio in comparing vaccine uptake to several determinants.

Determinants	n. of Studies	OR (95%CI)	I^2^	*p*-Value for Heterogeneity
Sex (male vs. female)	8	0.77 (0.62–0.96)	53%	0.040
Ethnic (caucasian vs. other)	4	1.91 (1.12–3.26)	81%	0.001
Disease (Crohn vs. UC)	8	0.92 (0.82–1.11)	0%	0.910
Therapy with steroids (yes/no)	2	0.51 (0.34–0.76)	0%	0.770
Therapy with immunosuppressant (yes/no)	2	1.16 (0.57–2.36)	0%	0.380
Therapy with biological drugs (yes/no)	5	0.90 (0.60–1.37)	74%	0.004
Marital status (married vs. not married)	2	1.07 (0.84–1.37)	0%	0.550
Previous diagnosis of COVID-19 (yes/no)	2	0.44 (0.29–0.65)	0%	0.340
Previous influenza vaccine uptake (yes/no)	2	5.01 (3.64–6.90)	0%	0.830

## Data Availability

Not applicable.

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
