# Peer review of "COVID-19 Immunization Rates in Patients with Inflammatory Bowel Disease Worldwide: A Systematic Review and Meta-Analysis"

_vaccines, 2023, doi:10.3390/vaccines11101523_

Round 1
Reviewer 1 Report
Thank you for sharing your manuscript. Here some comments that could help to improve the review article:
L96-98: The aim stated here is not well in line with the title of your manuscript. Please revise your title to better reflect the overarching aim of your manuscript to potential readers.
General comment: As this aims to be a rather global review, it is mandatory to include the vaccine type(s) licensed in the respective countries and actually administered to be able to correctly assess objectives of your manuscript such as vaccine coverage, vaccine uptake and vaccine attitudes including vaccine hesitancy.
Please see above.
Author Response
Q1. L96-98: The aim stated here is not well in line with the title of your manuscript. Please revise your title to better reflect the overarching aim of your manuscript to potential readers.
A1. Revised.
Q2. General comment: As this aims to be a rather global review, it is mandatory to include the vaccine type(s) licensed in the respective countries and actually administered to be able to correctly assess objectives of your manuscript such as vaccine coverage, vaccine uptake and vaccine attitudes including vaccine hesitancy.
A2. We implemented table 1 with the requested info.
Reviewer 2 Report
In the present meta-analysis Bianchi et al summarized the main data about COVID-19 vaccination in IBD patients: coverage, uptake and hesitancy. Main comments:
1) A linguistic revision is necessary (see for example “confronted” page 1 line 12).
2) Page 2 line 76: Beizzo - - > Bezzio.
3) The title does not reflect the content of the article (complication risk was not assessed), therefore it should be re-written.
4) Please explain if fixed or random effects model was used.
5) Based on which assumption the cutoff of 419 patients was chosen to discriminate between large and small studies?
6) Regarding the results reported in section 3.4, literature search was not satisfactory, as some studies have not been included, e.g. Principi M et al, Eur J Gastroenterol Hepatol 2023.
See above
Author Response
Q1. A linguistic revision is necessary (see for example “confronted” page 1 line 12).
A1. The manuscript has been revised by a native speaker.
Q2. Page 2 line 76: Beizzo - - > Bezzio.
A2. Revised.
Q3. The title does not reflect the content of the article (complication risk was not assessed), therefore it should be re-written.
A3. Revised.
Q4. Please explain if fixed or random effects model was used.
A4. We used a random effect model. We implemented the methods section.
Q5. Based on which assumption the cutoff of 419 patients was chosen to discriminate between large and small studies?
A5. As reported in methods section, we used a statistical criterion to define this cut-off (the median of the sample size of the included studies). Another cut-off usually used in meta-analysis is 1,000+ vs. <1,000 subjects, but in this case we considered more fit the statistical criterion based on median value.
Q6. Regarding the results reported in section 3.4, literature search was not satisfactory, as some studies have not been included, e.g. Principi M et al, Eur J Gastroenterol Hepatol 2023.
A6. We can assure you that the systematic search was conducted meticulously, following a rigorous methodology. The inclusion and exclusion criteria were applied as outlined in the methods section. Regarding the study by Principi M et al., it primarily centered on a group of vaccinated subjects, evaluating the vaccine's safety profile and factors contributing to vaccine hesitancy. As a result, it did not provide relevant insights into vaccine coverage, which is the central focus of our paper. Therefore, we opted to exclude it from our analysis. Nonetheless, we have incorporated relevant findings from that study into the discussion section, considering its implications for the broader discussion of vaccines in the context of IBD patients.
Round 2
Reviewer 1 Report
Thank you for addressing my comments at least in part. The vaccine type(s) licensed in the respective countries are stated in Table 1. However, to make your research more meaningful, the vaccine(s) actually administered should be looked at to assess variables inclusive of vaccine coverage, vaccine uptake and vaccine attitudes including vaccine hesitancy.
Please see above.
Author Response
Regrettably, the majority of the included studies did not provide information on the specific vaccine types used, and in cases where such information was reported, it often lacked granularity. Consequently, we are unable to fulfill your request comprehensively. We have acknowledged this limitation in the study's dedicated section on limitations.
Reviewer 2 Report
Answers were fine.
Author Response
Thank you